# Real-time imaging of cellular forces using optical interference

Andrew T. Meek[1], Nils M. Kronenberg [1,2], Andrew Morton[1], Philipp Liehm [1], Jan Murawski[1], Eleni Dalaka [1], Jonathan H. Booth[1], Simon J. Powis [3,4] & Malte C. Gather [1,2,4 ✉]

Important dynamic processes in mechanobiology remain elusive due to a lack of tools to image the small cellular forces at play with sufficient speed and throughput. Here, we introduce a fast, interference-based force imaging method that uses the illumination of an elastic deformable microcavity with two rapidly alternating wavelengths to map forces. We show real-time acquisition and processing of data, obtain images of mechanical activity while scanning across a cell culture, and investigate sub-second fluctuations of the piconewton forces exerted by macrophage podosomes. We also demonstrate force imaging of beating neonatal cardiomyocytes at 100 fps which reveals mechanical aspects of spontaneous oscillatory contraction waves in between the main contraction cycles. These examples illustrate the wider potential of our technique for monitoring cellular forces with high throughput and excellent temporal resolution.

[1] SUPA, School of Physics and Astronomy, University of St Andrews, North Haugh, St Andrews, UK. [2] Humboldt Centre for Nano- and Biophotonics, Department of Chemistry, University of Cologne, Cologne, Germany. [3] School of Medicine, University of St Andrews, North Haugh, St Andrews, UK. [4] Centre of Biophotonics, University of St Andrews, North Haugh, St Andrews, UK. ✉email: mcg6@st-andrews.ac.uk

The interference of light allows exceptionally precise measurements of relative displacements, with applications ranging from the detection of gravitational waves to spectroscopy and optical telecommunications. In biomedical imaging, optical interference is used to obtain qualitative contrast (e.g., phase and differential interference contrast)[1,2], optical sectioning (e.g., optical coherence tomography)[3] and quantitative phase or topography information[4–6]. Mechanobiology, the application and sensing of forces by cells, is at the heart of many important phenomena in biology[7,8]. Measuring the pico- to nanonewton forces involved requires accurate mapping of the small deformations they induce, rendering interference a prime candidate in principle. However, the currently most widely used methods rely instead on localisation microscopy (e.g., elastic micropillars[9,10], traction force microscopy[11–13]) and spectroscopic imaging (e.g., Förster resonance energy transfer-based tension sensors[14–16]). Although these techniques allow the study of a number of mechanical processes in great detail, many important aspects remain elusive because spatially resolved maps of the small forces at play can often not be acquired with adequate speed, throughput and resolution. This is in part due to the need to acquire zero-force reference images, to use phototoxic light levels, and to perform computationally intensive offline data analysis. We recently introduced an interference-based approach, elastic resonator interference stress microscopy (ERISM), which offers reference-free, high-precision force measurements with μm lateral resolution and low probe-light intensity[17,18]. However, the use of ERISM has been limited to studying phenomena on the second timescale and longer[19,20,21], as each ERISM measurement requires the acquisition of at least 50 images at different illumination wavelengths which limits the acquisition speed to 0.4 fps; this is followed by data analysis that takes ~10 s per image.

Here, we introduce wavelength-alternating resonant pressure microscopy (WARP), a different interferometric approach that reaches sustained force imaging speeds of 100 fps and offers real-time operation rather than involving offline data analysis. WARP maintains the reference-free operation, piconewton precision, high lateral resolution and low light exposure of ERISM and thus enables prolonged and continuous force imaging at a high frame rate without affecting the cells under investigation. To demonstrate the capability of WARP, we show live imaging of the piconewton pushing forces associated with podosome protrusion in primary human macrophages. In addition, the unique combination of high frame rate and exquisite force sensitivity allows the detection of previously hidden forces, e.g., in cardiomyocytes where we observed millisecond, nanometre deformations induced by spontaneous oscillatory contraction (SPOC) waves flowing through sarcomeres in between the main contractions.

## Results

### Wavelength-alternating interference imaging of a microcavity.
Our method relies on the interference of light within a low finesse elastic microcavity consisting of a silicone-based elastomer sandwiched between two semi-transparent gold layers (Fig. 1a). When cells cultured on the surface of the cavity exert forces, they deform the top mirror, thereby inducing a local spectral shift in the cavity resonances. Imaging the reflected light intensity from the cavity under widefield, monochromatic illumination with two rapidly alternating wavelengths creates two related interference patterns, from which we can accurately compute the local cavity displacement as is explained in the following. Once the local cavity displacement is known, finite element modelling (FEM) can be applied[17] to compute a map of the mechanical stress that cells exert on the microcavity chip.

For a low finesse cavity filled with a medium of refractive index $n$, the intensity $I$ of light reflected from a point on its surface is a function of local cavity thickness $d$, and probe wavelength $\lambda$, and can be approximated by a cosine[22] with an offset $B$ and amplitude $A$ as:

$$I(\phi) \sim B + A\cos(2\phi) \qquad (1)$$

where

$$\phi = \frac{2\pi nd}{\lambda}. \qquad (2)$$

As the cosine function is symmetric, the direction of cavity deformations cannot be determined from Eq. (1) alone, and thus the direction of forces exerted on the cavity cannot be established from imaging the reflection from the cavity under illumination with only one wavelength. In practice, $A$ and $B$ also vary across the cavity surface; steep gradients in cavity thickness forming in areas of localised or particularly strong cellular force reduce the oscillation amplitude owing to spatial averaging, whereas inhomogeneities in illumination across the field of view lead to changes in the background term (see Supplementary Note 1 for an example).

Our method resolves the ambiguity in force direction and the issue of amplitude variation by taking two separate images of the cavity reflected intensity under illumination with two slightly different wavelengths (Fig. 1b–c). Subtracting these two images from each other removes the background term at each pixel while preserving a sinusoidal interference term at an intermediate frequency, and producing a slowly varying envelope sine function:

$$I(\lambda_1) - I(\lambda_2) = \left(B + A\cos\frac{4\pi nd}{\lambda_1}\right) - \left(B + A\cos\frac{4\pi nd}{\lambda_2}\right)$$
$$= -2A\sin\frac{2\pi n(\lambda_1 + \lambda_2)d}{\lambda_1\lambda_2}\sin\frac{2\pi n(\lambda_2 - \lambda_1)d}{\lambda_1\lambda_2} \quad (3)$$
$$= -2A\sin\phi_{\text{fast}}\sin\phi_{\text{slow}}$$

Addition of the two images instead produces cosine terms with the same frequencies:

$$I(\lambda_1) + I(\lambda_2) = \left(B + A\cos\frac{4\pi nd}{\lambda_1}\right) + \left(B + A\cos\frac{4\pi nd}{\lambda_2}\right)$$
$$= 2B + 2A\cos\frac{2\pi n(\lambda_1 + \lambda_2)d}{\lambda_1\lambda_2}\cos\frac{2\pi n(\lambda_2 - \lambda_1)d}{\lambda_1\lambda_2}$$
$$= 2B + 2A\cos\phi_{\text{fast}}\cos\phi_{\text{slow}}$$
$$\qquad (4)$$

This approach works because although the amplitude and background may vary across the image, we found that their change with wavelength is negligible, at least within one free spectral range of the cavity (18 nm for $\lambda \approx 630$ nm, $n \approx 1.40$ and $d \approx 8$ μm). The sum image still contains a background term ($2B$), but in practice, this background has negligible variations over the size of individual cells, and thus can be subtracted as shown later.

Dividing the intermediate background-subtracted sum and difference images by each other produces a final resultant image, where pixel intensity is expected to vary with cavity thickness according to:

$$\frac{I(\lambda_1) + I(\lambda_2) - 2B}{I(\lambda_1) - I(\lambda_2)} = \frac{\cos\phi_{\text{fast}}\cos\phi_{\text{slow}}}{-\sin\phi_{\text{fast}}\sin\phi_{\text{slow}}} \qquad (5)$$
$$= -\cot\!an\phi_{\text{fast}}\cot\!an\phi_{\text{slow}}$$

As the cotangent is asymmetrical, Eq. (5) now indicates the direction of thickness changes and hence the direction of applied forces (Fig. 1d–e). In addition, $A$ and $B$ have been removed. The

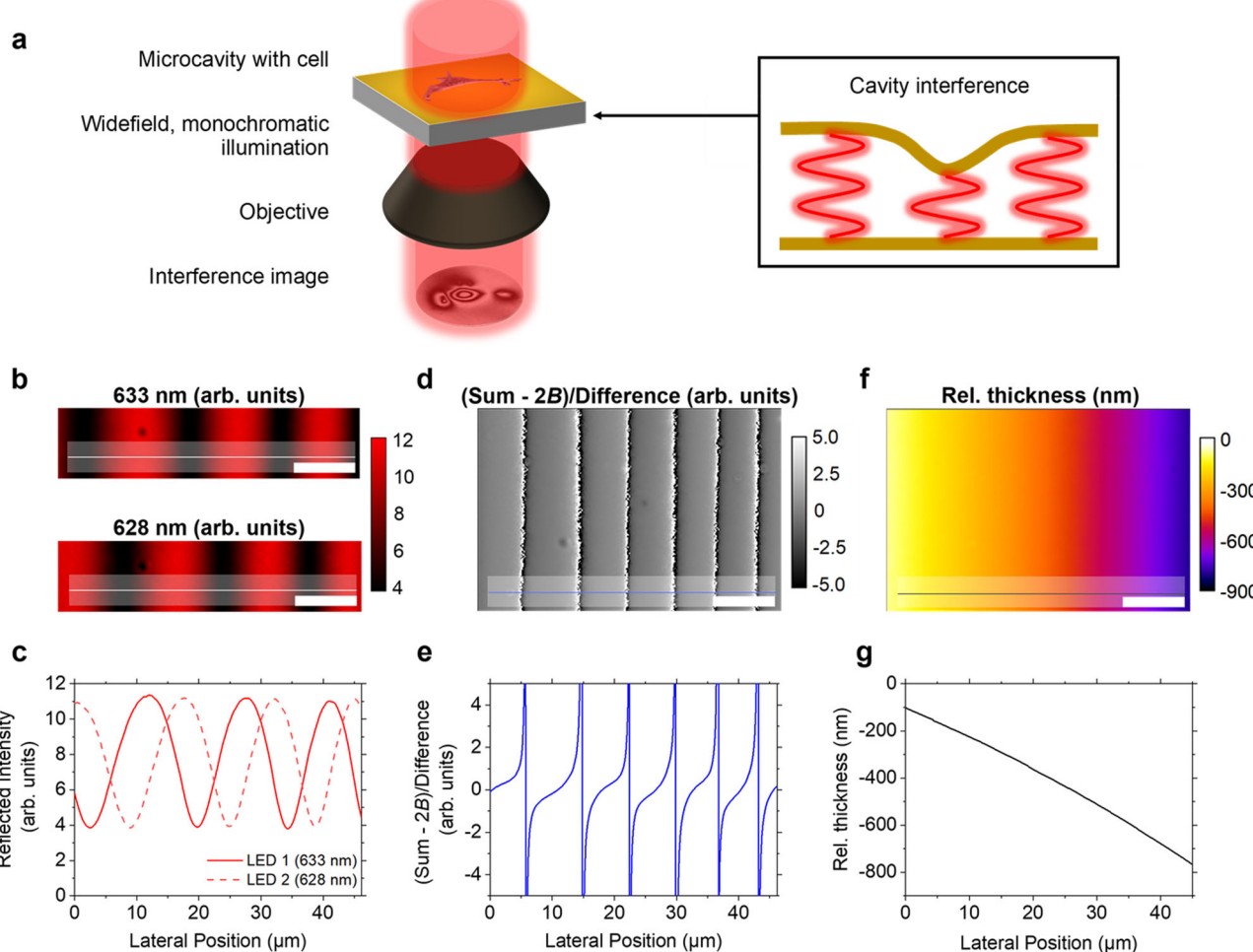

**Fig. 1 Principle of wavelength-alternating resonant pressure (WARP) microscopy. a** Schematic of elastic microcavity chip with a cell cultured on the top mirror surface (not to scale). As the cell exerts force, the cavity thickness changes locally, thus creating an interference fringe pattern when imaging the reflection from the cavity. **b** Interference fringes formed along a region of continuously varying cavity thickness for two different illumination wavelengths (633 nm and 628 nm). Colour bar in arbitrary units. **c** Intensity profiles along lines in **b** showing the phase difference between the two wavelengths. **d** Division image created by dividing the background-subtracted sum of the images in **b** by the difference of the images in **b**. Greyscale in arbitrary units. **e** Profile along the line in **d**, illustrating the presence of directional information in this new image that is absent from the original images in **b**. **f** Ground truth of relative cavity thickness for region shown in **d**, as obtained by wavelength-scanning ERISM. **g** Profile along line in **f**. A comparison with the profile in **e** yields a lookup table for relative thickness changes between fringes. Combined with automated fringe counting, this table links the numerical value at each point in a division image to the local displacement of the cavity top mirror. Scale bars, 10 μm.

image resulting from the division can be translated into a map of local relative thickness using a combination of (i) automated fringe counting, considering that the thickness difference between adjacent fringes is ≈ 113 nm, and (ii) a lookup table that converts the numerical value of Eq. (5) into a thickness change for the regions between fringe boundaries. This is done by opening both the division and difference images in a software script and scanning across the same line of pixels in parallel. Zero-crossing points in the difference image are identified as fringe transitions and, therefore, as transitions between adjacent fringe modes. The corresponding sign transition in the division image (positive-to-negative or negative-to-positive) determines the direction of displacement between fringes. For each microcavity with a specific combination of top and bottom mirror reflectance, there is a combination of uniquely shaped cotangent image modes, Mode 1 and Mode 2 (see Supplementary Note 2 on the algorithm for more details). Whether a pixel in the division image corresponds to Mode 1 or Mode 2 is established by considering the sign of the same pixel in the difference image. Establishing fringe transitions provides displacement accuracy to within a

single mode, i.e., to $d_{mode} \approx 113$ nm for wavelengths of 633 nm and 628 nm, or $d_{mode} \approx 125$ nm for wavelengths of 695 nm and 700 nm (see Fig. 2 and Supplementary Note 3). The fine nanometre detail within this range is then provided by mapping the specific value of the division image to displacement values using the lookup table calibration. The equation used to assign a specific displacement is:

$$d = p \times d_{mode} - n \times d_{mode} + d_{lookup} + d_{offset} \qquad (6)$$

When scanning across a row of pixels from left to right in the division image, the leftmost pixel is taken as the baseline displacement $d_{offset}$, and all further pixels are compared to this point by counting positive and negative fringe transitions (an increase or decrease in displacement). When a fringe transition is negative-to positive, the multiplier $p$ is incremented, whereas if it is positive-to-negative, the integer $n$ is incremented; therefore, the appropriate number of mode changes ($d_{mode}$) is added and subtracted. Finally, the greyscale value in the division image is compared with a lookup table (created from a region of the cavity with gradually varying thickness or by comparing the division

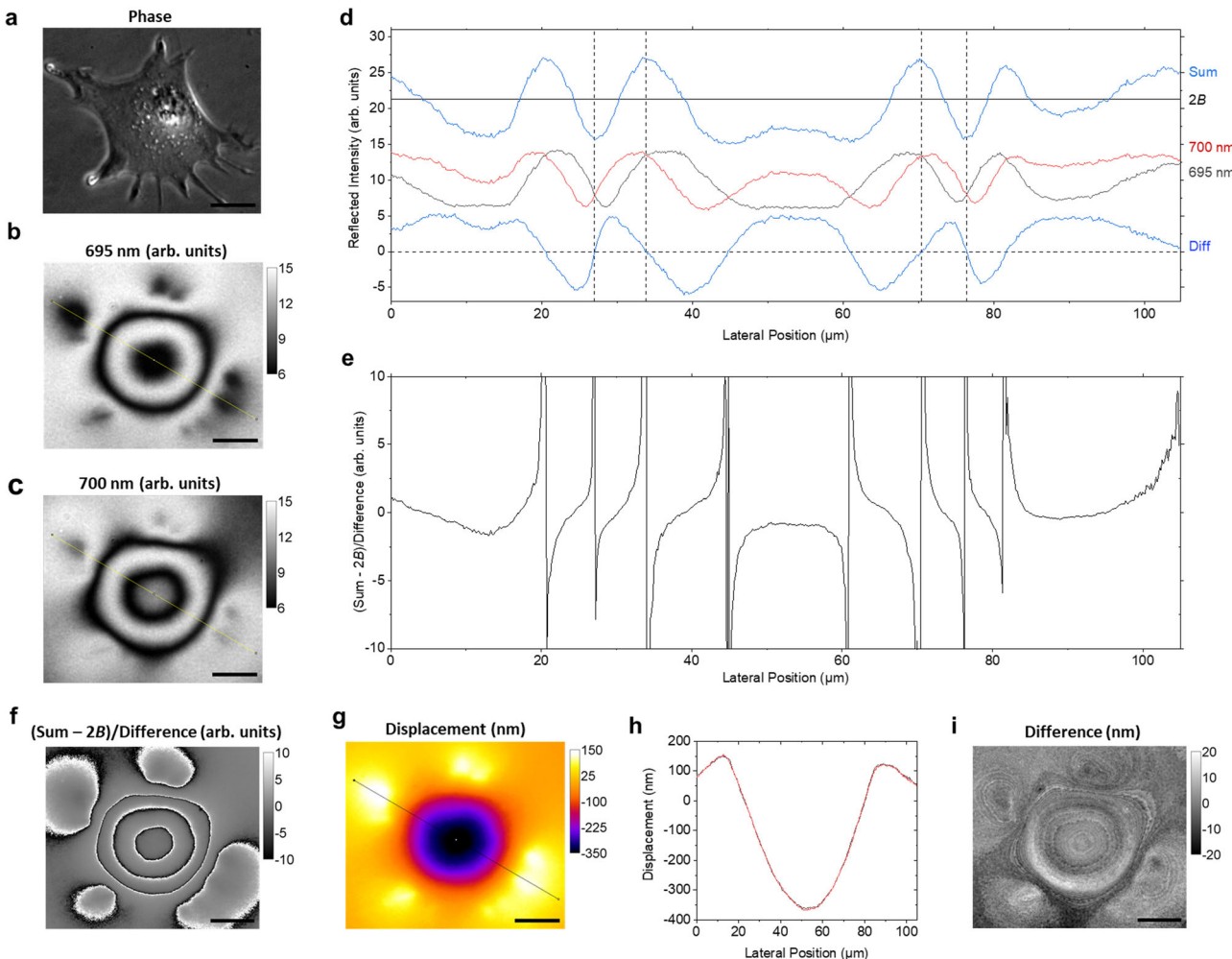

**Fig. 2 Testing the accuracy of WARP microscopy for NIH-3T3 cells. a** Phase-contrast image of an NIH-3T3 cell. **b, c** Monochromatic interference images of the same cell at 695 nm and 700 nm, respectively. Greyscale in arbitrary units. **d** Plot along the yellow line in the monochromatic images in **b** and **c**. A plot along the same line for the sum and difference images is also shown, with the match between zero-crossing points and maxima/minima used to establish the background. **e** Ratio of the sum and difference profile in **d**, after subtracting the background 2B from the sum. **f** Division image corresponding to the profile in **e**. **g** Displacement map generated from the division image via automated fringe counting and use of a lookup table. **h** Profile plot along the black diagonal line in **g** (black) and profile plot along the same line for a displacement map generated by wavelength-scanning ERISM (red). The black line is mostly hidden behind the red line due to the good agreement between both approaches. **i** Difference between the displacement map in **g** and the corresponding displacement map obtained by wavelength-scanning ERISM. Scale bars, 20 μm. Images are for a representative cell from three independent experiments.

image to the absolute cavity thickness as obtained via our previously developed wavelength-scanning ERISM method), and the corresponding fine displacement $d_{\text{lookup}}$ within the range of the mode is added to the broad displacement obtained from the previous counting of mode transitions. Before scanning horizontally along each row of pixels, a single scan is made down the first vertical column of pixels to establish the baseline displacement $d_{\text{offset}}$ for each row. Finally, the mean displacement offset for the image is calculated across an area of uniform cavity thickness and subtracted from the final image to create the zero-displacement baseline.

For a given cavity and a given set of acquisition conditions, the lookup table only needs to be created once, and this can be done in a region of the cavity that does not contain cells but shows a gradual variation in thickness (Fig. 1f–g, gradual thickness variations occur naturally at the edge of the sensor due to the spin-coating based deposition of the elastic material in the cavity, Methods). A more detailed discussion of the

algorithm underlying WARP is given in Supplementary Note 2, where we discuss the more accurate Fabry Perot resonator equation[23] instead of the simplified Eq. (1) and describe how potential ambiguities can be avoided.

To implement the above scheme, light from two intensity matched light-emitting diodes (LEDs) was passed through narrowband transmission filters that were set to two different wavelengths by angular tuning (e.g., 628 nm and 633 nm centre wavelength with 1 nm bandwidth), combined by a beamsplitter and then coupled into an inverted microscope. A fast scientific CMOS camera imaged the reflection from the cavity while the illumination wavelength was alternated between each acquired frame (Methods). Our setup also included a scanning monochromator and a halogen lamp to support our previously reported ERISM method that scans through a range of illumination wavelengths[17]. This enabled a direct comparison of the two methods, as both can use the same elastic microcavity chip.

**Accuracy of wavelength-alternating interference imaging**. We first established the accuracy of the wavelength-alternating imaging approach for the static deformation induced by NIH-3T3 cells. This was achieved by computing the deformation patterns from two images taken at two different wavelengths and comparing these patterns to those obtained by the slower ERISM approach, using a set of 201 images taken at illumination wavelengths varying in 1 nm steps from 550 to 750 nm (Supplementary Note 3). Figure 2a shows a phase-contrast image of an NIH-3T3 cell attached to a microcavity chip, whereas Fig. 2b and c show the reflection of the chip for the same field of view under monochromatic illumination with 695 nm and 700 nm light, respectively.

Figure 2d shows a profile plot along the same lines in the two monochromatic images, along with the sum and difference of both profiles. The zero points in the difference image were used to establish the location of the maxima and minima in the sum image and thus to accurately determine the background term $2B$. Dividing the background-subtracted sum profile by the difference profile yields the plot in Fig. 2e, which resembles the behaviour of the cotangent function in Eq. (5). Performing the same division on the entire images gives Fig. 2f, which was then converted to a displacement map using fringe counting and a lookup table as discussed before (Fig. 2g).

Having produced the displacement map using the alternating-wavelength approach, we also obtained a ground truth displacement map with the conventional ERISM procedure. Visually both maps looked identical (Supplementary Note 3). Profiles through the centre of the maps also show excellent agreement between them (Fig. 2h). The agreement is corroborated further by a difference image of the two maps (Fig. 2i). The histogram of this difference image has a full width at half maximum of 9.4 nm, corresponding to 2% of the total displacement caused by the cell (Supplementary Note 3).

**Real-time mapping of fibroblasts and macrophage podosomes**. To demonstrate the real-time capability of our method, we continuously acquired wavelength-alternating images while scanning across a culture of NIH-3T3 cells on a cavity chip by moving the microscope stage. We then generated live maps of cell-induced deformation at a frame rate of 10.5 fps by performing real-time image subtraction, addition, division, fringe counting and mapping to the lookup table (Fig. 3a–c, Supplementary Videos 1 and 2).

To show that WARP can resolve forces down to a few piconewtons, we also performed real-time imaging of podosomes of human macrophages[24,25]. Here, a spatial Fourier filter was applied to the original displacement maps to visualise the activity of the μm-sized podosomes more clearly against the background of broad cell-wide deformations (Fig. 3d). In addition, displacement maps were converted into stress maps using FEM (Fig. 3e; FEM is too computationally expensive to perform in real-time, but the resultant stress maps visually resemble the displacement maps, so a real-time assessment of stress can be made from the latter). Individual podosomes exerted stresses between <1 Pa and ~15 Pa over areas of typically 0.1–5 μm$^2$, which corresponds to a podosomal force of between 0.1 and 20 pN (Fig. 3e–h, Supplementary Video 3). As a consistency check, we compared the podosome forces determined from the stress maps generated by FEM to the forces estimated by a simple Hertz model that approximates each podosome as a cylindrical indenter and taking indentation depth and podosome size from the deformation map. The values obtained with both methods showed close agreement (Supplementary Note 4). In addition, we used atomic force microscopy (AFM) to indent into the microcavity with a set of

known forces and compared these applied forces to the forces calculated from the WARP displacement maps using our FEM routine. On average the deviations between the force applied by AFM and the force computed via WARP were <10% (Supplementary Note 5).

The data also showed that podosomes are highly dynamic, with oscillations in exerted force, podosome movement and formation of new podosomes. Consistent with earlier reports in the literature[17,25,26], we observe characteristic oscillations in force over tens of seconds. In addition, podosome force also showed pN fluctuations on even shorter timescales (Fig. 3g; see Supplementary Note 6 for frequency analysis of data, a control measurement in a region not containing a podosome, and an image indicating the analysed regions in the stress map). In a different study, Labernadie et al.[25] also observed fluctuations in podosome protrusion on similar timescales, using AFM on the backside of thin membranes to which macrophages adhered. Although the depth of podosome protrusion was comparable in both studies, podosomal forces differed significantly, with the other work finding forces in the nN range. We attribute this difference to the fact that cells were cultured on substrates that differed in compliance by many orders of magnitude (WARP microcavity chips, ≈5 kPa; protrusion force microscopy, ≈2 GPa in a 60 nm thick membrane). As the vertical depth of the protrusions achieved by podosomes is very similar on the two different substrates, a stark difference in the force required to achieve these protrusions is expected, and the results reported by Labernadie et al. are thus not in contradiction to our findings.

Our data also allowed analysis of the area over which podosomes protrude, and to compare this to the exerted force in a high-throughput manner (Fig. 3h). For podosomes indenting over areas larger than 0.7 μm$^2$, we observed a strong linear correlation between force and area in line with earlier findings based on point-scanning protrusion force microscopy[25]. For podosomes with footprints <0.7 μm$^2$, the data deviates from the linear trend. The size of these podosome footprints becomes comparable to the optical resolution of the used microscope objective (numerical aperture 0.6; $r_{Airy} \approx 640$ nm) and so these likely appear larger because their actual footprint is convolved with an airy diffraction pattern.

**High frame rate imaging of cardiomyocyte contraction**. Besides offering real-time capability, our method allows imaging of rapidly changing force patterns, as are generated for example by beating cardiomyocytes. To follow the deformations induced by cardiomyocyte contractions, we increased the acquisition rate of our method further—now acquiring WARP maps at 100 fps (Fig. 4a, b, Supplementary Video 4). Here, we used offline image analysis; we expect that real-time analysis would be feasible through the use of dedicated GPU hardware[27]. Unlike the podosomes analysed above, cardiomyocyte contractions primarily exert horizontal forces on the substrate. We have previously shown that horizontal forces can be quantified by measuring the magnitude of local twisting they induce to the microcavity surface, and that focal adhesion forces measured in this way are consistent with other literature reports[17]. However, this procedure has required manual image analysis and is thus time-consuming for the multi-frame data sets generated by WARP. To quickly quantify the total force exerted by a cardiomyocyte and the temporal evolution of this force, we instead computed the volume by which the cell indents into the substrate for each frame. This analysis revealed rapid cycles of contraction and relaxation as well as small gradual changes in force amplitude (Fig. 4c, d). To illustrate the compatibility of our method with biochemical assays and to validate that we obtain a meaningful mechanical readout, cardiomyocytes

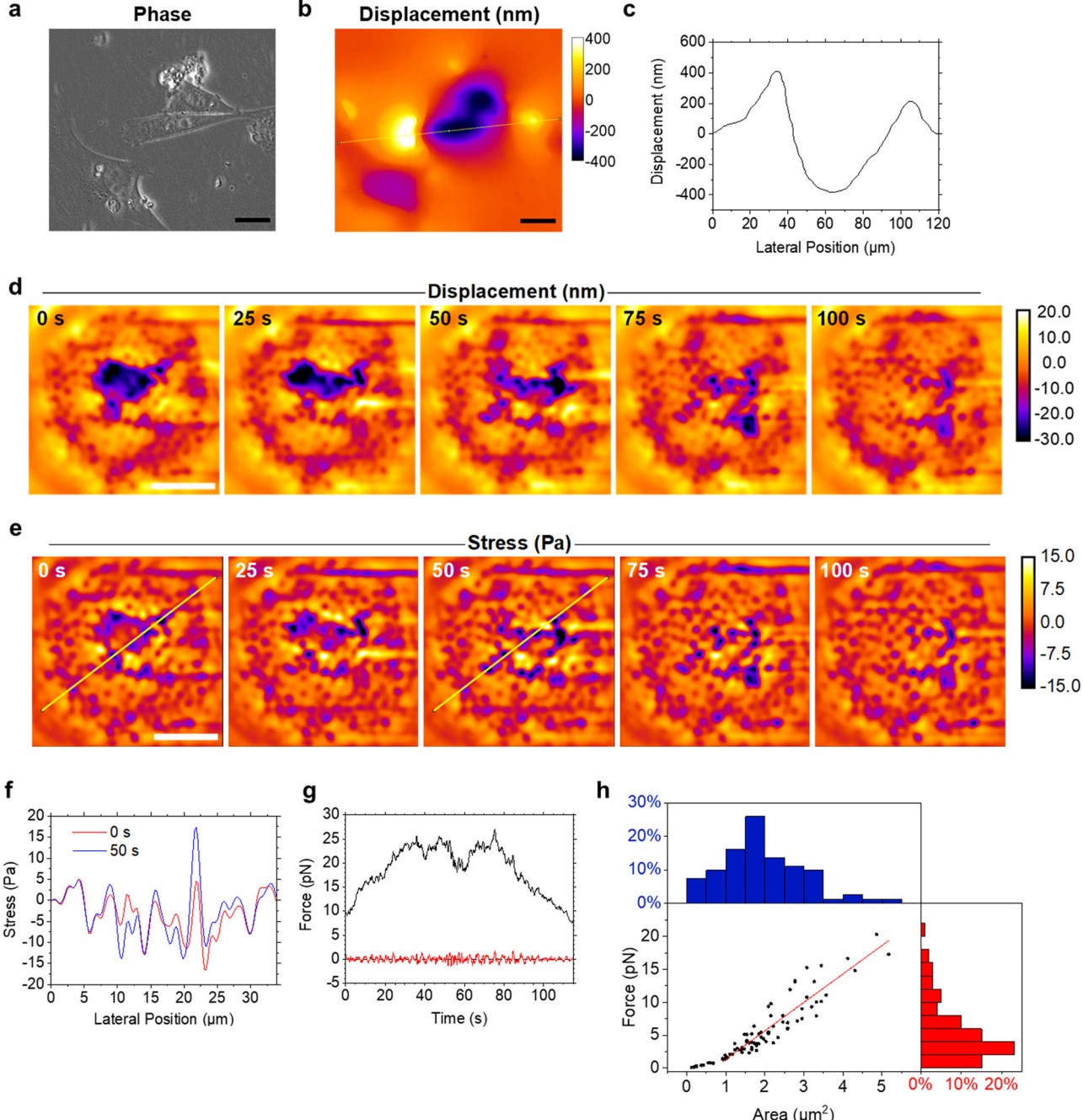

**Fig. 3 Real-time imaging of cellular forces. a** Phase-contrast microscopy image of NIH-3T3 cells on the surface of a microcavity chip. **b** Displacement map for the same field of view as in **a**, generated in real-time while scanning across the surface of the chip by moving the microscope stage. **c** Profile along the yellow line in **b**. **d** Selected frames from live force imaging of a macrophage at 17 fps, showing the microcavity displacement caused by podosome protrusions. Spatial Fourier-filtering was applied to emphasise podosome protrusions against the background of broad indentation generated by the cell. **e** Maps of the stress exerted by the podosome protrusions as obtained from the displacement time-lapse in **d** using FEM model analysis. **f** Plot along the line in **e** at two different time points, showing the fluctuations of podosome stress. **g** Force exerted by a typical podosome over time (black, filtered with 0.5 s moving average). Applying an additional high pass temporal Fourier filter (cutoff frequency, 0.2 Hz) removes the slowly varying features and resolves the fast pN fluctuations in force more clearly (red). **h** Analysis of correlation between podosomal force and area of indentation for $n = 81$ podosomes (black symbols). Red line indicates linear fit to the data ($R^2 = 0.85$) for podosome areas >0.7 $\mu m^2$. Histograms of podosomal forces (red) and area of indentation (blue) for all podosomes analysed in the main plot. Scale bars, 20 $\mu m$ **a**, **b**; 10 $\mu m$ **d**, **e**. Images in **a** and **b** are representative of 20 independent observations from one experiment. Images in **d** and **e** are representative of three independent observations from one experiment.

were challenged with the calcium channel inhibitor nifedipine. Consistent with previous studies[28], this resulted in an immediate and substantial reduction in overall cell force and beating activity (80–96% reduction in amplitude and 20% reduction in baseline; Fig. 4c, e–g, Supplementary Note 7). The reduction in force correlated with a 44% increase in contraction frequency, which agrees with previous work involving stem cell-derived cardiomyocytes[29]. In addition, the coefficient of variation of both the contraction force and frequency increased, indicating less regular beating behaviour (insets to Fig. 4f and g). The original

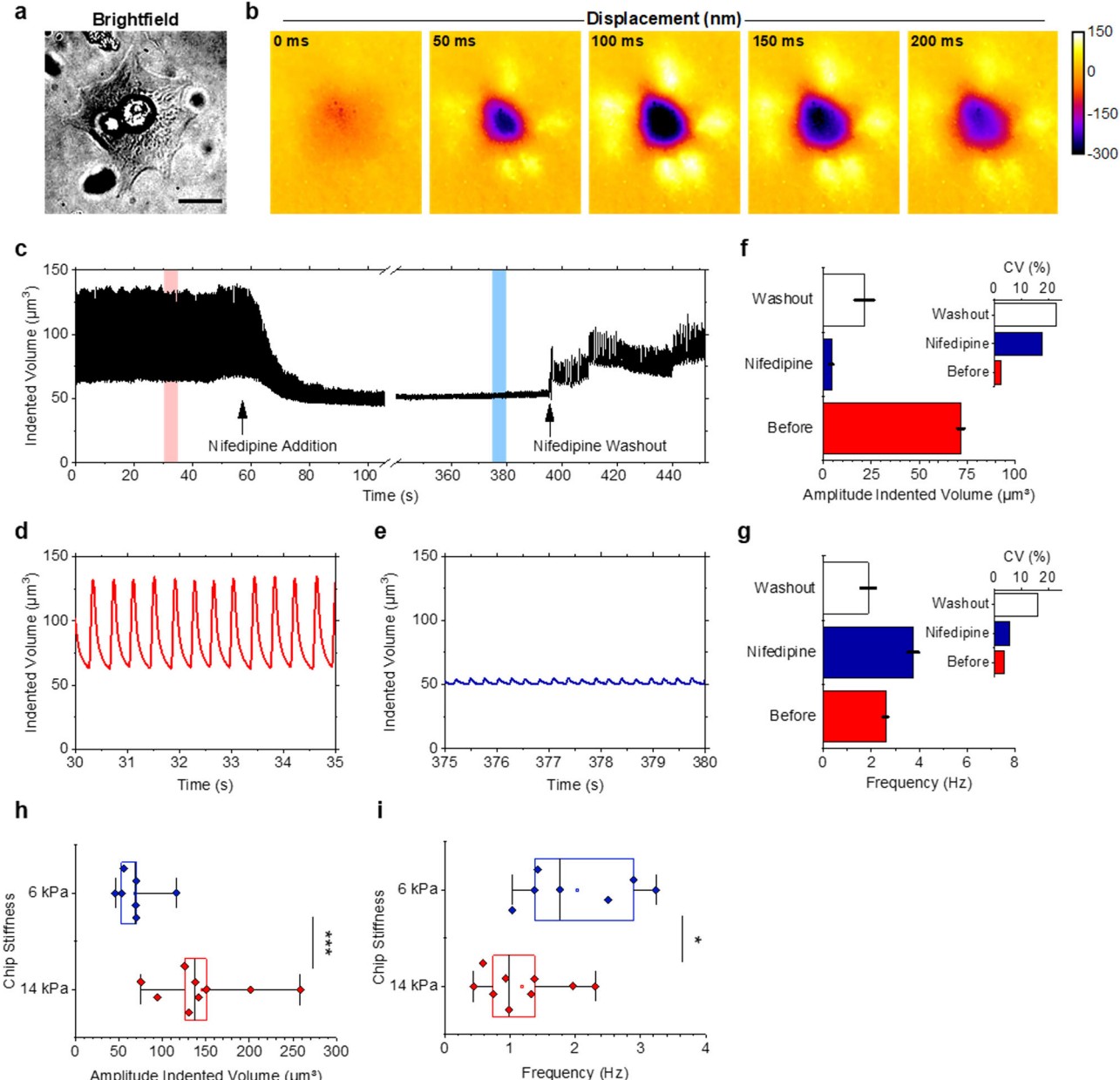

**Fig. 4 High-speed contraction imaging of beating cardiomyocytes. a** Brightfield image and **b** section from a 100 fps time-lapse of displacement images for a periodically contracting neonatal cardiomyocyte. **c** Temporal evolution of total force exerted by the cell, as quantified by the volume it indents into the microcavity chip, before, and upon administration of the calcium channel inhibitor nifedipine and following a subsequent washout. Data extracted from a 7-min time-lapse containing a total of 43,000 frames. **d** Data from interval marked by the red box in **c** on a finer timescale showing the periodic contractions before the addition of nifedipine. **e** Data from interval marked by blue box in **c** indicating a significant reduction in amplitude of contraction force after addition of nifedipine. **f** Amplitude of contraction forces over 20 s intervals before, during and after nifedipine exposure. Data shows the mean ±standard deviation (s.d.). Inset, coefficient of variation (CV) for each case. $n = 52$ contractions for before, $n = 76$ contractions for nifedipine, $n = 36$ contractions for washout. **g** Frequency of contraction over 20 s intervals before, during and after nifedipine exposure. Data show the mean ±s.d. Inset, CV for each case. $n = 52$ contractions for before, $n = 76$ contractions for nifedipine, $n = 36$ contractions for washout. **h** Box plot of contraction force over 20 s interval for two chips with different apparent stiffness. $n = 9$ cells for 14 kPa, $n = 7$ cells for 6 kPa. One-tailed Mann–Whitney $U$ test, $P < 0.001$ (***$P = 0.00035$). **i** Box plot of contraction frequency for data in **h**. $n = 9$ cells for 14 Pa, $n = 7$ cells for 6 kPa. One-tailed Mann–Whitney $U$ test, $P < 0.05$ (*$P = 0.013$). Line, box and whiskers in **h** and **i** show median, interquartile range (i.e., difference between the 75th and 25th percentiles) and maximum/minimum data points, respectively. Scale bar, 20 μm. Images are representative of two independent cardiomyocyte experiments.

beating phenotype can be partially recovered by nifedipine washout, but the fluctuation in force amplitude and frequency remained substantial.

Using a similar protocol, we also investigated the effect of substrate stiffness on cardiomyocyte force and contraction frequency. Cardiac tissue has been measured to have Young's modulus in the range of 10–20 kPa[30,31]. Cardiomyocytes were, therefore, cultured on two microcavity chips with different apparent stiffnesses (14 kPa and 6 kPa, i.e., similar to and softer than physiological conditions, respectively). Cells on the 14 kPa chip showed significantly stronger contraction forces, with the indented volumes being larger despite the stiffer surface (Fig. 4h).

These results align with earlier findings from TFM[30] and high-speed line scanning imaging of micropillar arrays[32]. Varying the surface stiffness of the microcavity chip also affected contraction frequency, with higher contraction frequencies observed on the softer chip (Fig. 4i). For both stiffnesses, we observe a negative correlation between the contraction force and contraction frequency of individual cells, with stronger cells showing smaller contraction frequency (Supplementary Note 8). These findings corroborate data from prior work that investigated the effect of surface stiffness on the beating frequency for embryonic cardiomyocytes[33]; experiments with stem cell-derived cardiomyocytes[34] also indicated an inverse relationship between contraction amplitude and contraction frequency.

**Spontaneous micro-contractions and long-term measurements**. In addition to confirming previously observed mechanical characteristics of cardiomyocytes, the nm-displacement resolution and large dynamic range of WARP were able to resolve additional complex force patterns that were associated with spontaneous micro-contractions that some cardiomyocytes performed during their resting phase (Fig. 5a, b, Supplementary Video 5). These micro-contractions typically lasted for <100 ms and caused deformations <30 nm, meaning they would be difficult or impossible to observe with other force mapping techniques. We attribute these contractions to SPOCs, which represent a characteristic behaviour of cardiomyocytes between relaxation and contraction during which each sarcomere in a myofibril undergoes rapid length oscillations[35,36].

WARP uses low-intensity red light and can therefore also monitor the mechanical activity of cells continuously over long periods without inducing phototoxic effects. To demonstrate this, a cardiomyocyte cluster consisting of three cells was monitored continuously over 1 h at a frame rate of 100 fps. Figure 5c shows the indented volume in the central region of one cell in the cluster over the entire 1 h period (see Supplementary Note 9 and Supplementary Video 6 for further information). We then analysed the temporal variation in the contraction frequency and in the maximum indented volume during each contraction (Fig. 5d), and we found that the envelope of the contraction force mirrors the variation in contraction frequency. Over the entire 1 h measurement, there was a strong correlation between peak indented volume and contraction frequency for contraction frequencies up to 8 Hz (Fig. 5e). For contractions at >8 Hz frequency, this correlation broke down, but 98.9% of contractions had a frequency <8 Hz (histogram in Fig. 5e). We estimate that the 100 fps acquisition rate of WARP allows accurate assessment of contraction frequency and peak force up to frequencies of at least 20 Hz, sufficient to record the vast majority of physiological behaviours. The observed fluctuation in beating frequency is typical for single neonatal cardiomyocytes and for small clusters of cells as investigated here; the rhythm of contractions is known to stabilise once the number of cardiomyocytes in a network exceeds approximately eight cells[37].

## Discussion

We presented a new optical interference-based method for cellular force imaging that makes use of illumination with two alternating wavelengths to rapidly acquire widefield maps of cell-induced deformations. The technique evolved from ERISM[17], and foundational features such as the use of an elastic microcavity and high spatial (μm) and mechanical (pN) resolution are consistent between them. However, the speed of acquiring data and of computing deformation maps was improved dramatically by reducing the number of distinct wavelengths required to obtain a deformation map from at least fifty to just two.

WARP can perform data acquisition and analysis in parallel and in real-time. This allowed for the creation and display of live displacement maps with sustained frame rates of over 10 fps while freely scanning across cell cultures on the microscope stage to identify regions of interest. In addition, the nanoscale indentations and oscillations of podosomes formed by human macrophages were imaged at 17 fps, revealing μm and sub-μm-sized indentations, which exerted forces in the piconewton range. The forces fluctuated in the sub-piconewton range on short timescales (<5 s). In addition, a strong correlation was observed between the forces exerted by podosomes and the area over which they indent.

Performing the data analysis offline enabled continuous acquisition at 100 fps, which allowed us to image the contractions of beating neonatal murine cardiomyocytes over extended time periods. The high temporal and mechanical resolution of WARP revealed unique features of the cardiomyocytes on the millisecond timescale, that are likely associated with SPOC waves flowing through the cell during the resting phase of the contraction cycle. There is speculation that SPOC waves are involved in regulating the heartbeat by creating an intermediate phase between contraction and relaxation[35], and in the future, the ability to monitor the mechanical strength and evolution of SPOC waves may help elucidate this aspect further. Extending our measurements to the 1 hour timescale showed a robust correlation between peak contractile force and beating frequency.

In summary, WARP enables the mapping of nanoscale cellular forces in real-time, thus extending the study of cellular force into previously inaccessible realms. The technique provides excellent spatial and temporal resolution, is label-free, and due to the low light levels required, it involves minimal risk of phototoxicity. We expect that integration with high-resolution structural imaging, electrophysiology, and microfluidics is straightforward and will enable rapid cross-correlation between mechanics and other phenotypic information. Our method is thus well suited to complement existing high-throughput screening techniques, e.g., for cardiac safety testing in drug discovery[38,39].

## Methods

**Microscope**. The WARP setup was integrated into an inverted microscope (Nikon Eclipse Ti). A schematic of the setup and light path is shown in Supplementary Note 10. Collimated light from two identical red LEDs (625 nm centre wavelength, Thorlabs M625L3) was combined onto the same illumination path using a 50–50 beamsplitter. Narrowband filters centred at 632.8 nm with a FHWM of 1 nm (Thorlabs FL632.8-1) were used to select the desired measurement wavelengths from the LED spectra. A tilt of 15° was introduced to the filter at LED 2 to select 628 nm. The LED wavelengths were measured either with a fibre spectrometer, or by matching interference fringes to those produced under illumination with a monochromator coupled to a halogen light source. An achromatic doublet lens ($f = 150$ mm) focused the light from the two LEDs to the back focal plane of the microscope objective lens (Nikon S Plan Fluor ELWD 40×), generating a collimated beam incident perpendicular onto the microcavity chip. The resulting reflected interference image from the cavity was captured by an sCMOS camera (Andor Zyla 4.2 or Hamamatsu Orca Flash 4.0). A trigger signal from the camera and an IC 4017-decade counter circuit toggled between which LED was used for illumination.

**Data analysis**. Analysis of the acquired interference images was performed using ImageJ (Fiji package) and Python (Anaconda distribution). Alternate wavelengths were separated into individual stacks and the sum and difference taken between them, followed by the creation of a stack of division images. Images taken under illumination with LED 1 and LED 2 are temporally offset by the exposure time of the camera (e.g., ≈5 ms for the cardiomyocyte data). To mitigate motion artefacts due to this offset, images taken with LED 1 were compared to the average of the preceding and following image taken with LED 2. The stacks containing division and difference images were then opened using a Python script and the conversion to a displacement map was made through automated fringe counting and calibrated lookup tables (see Supplementary Note 2 and the main text for details). The conversion from the real-time displacement maps to stress maps was performed offline via FEM (COMSOL), using the procedure outlined in ref. [17]. In brief, using the known Young's modulus of the elastomer and the displacement maps allows the calculation of the Cauchy stress tensor, the vertical component of which

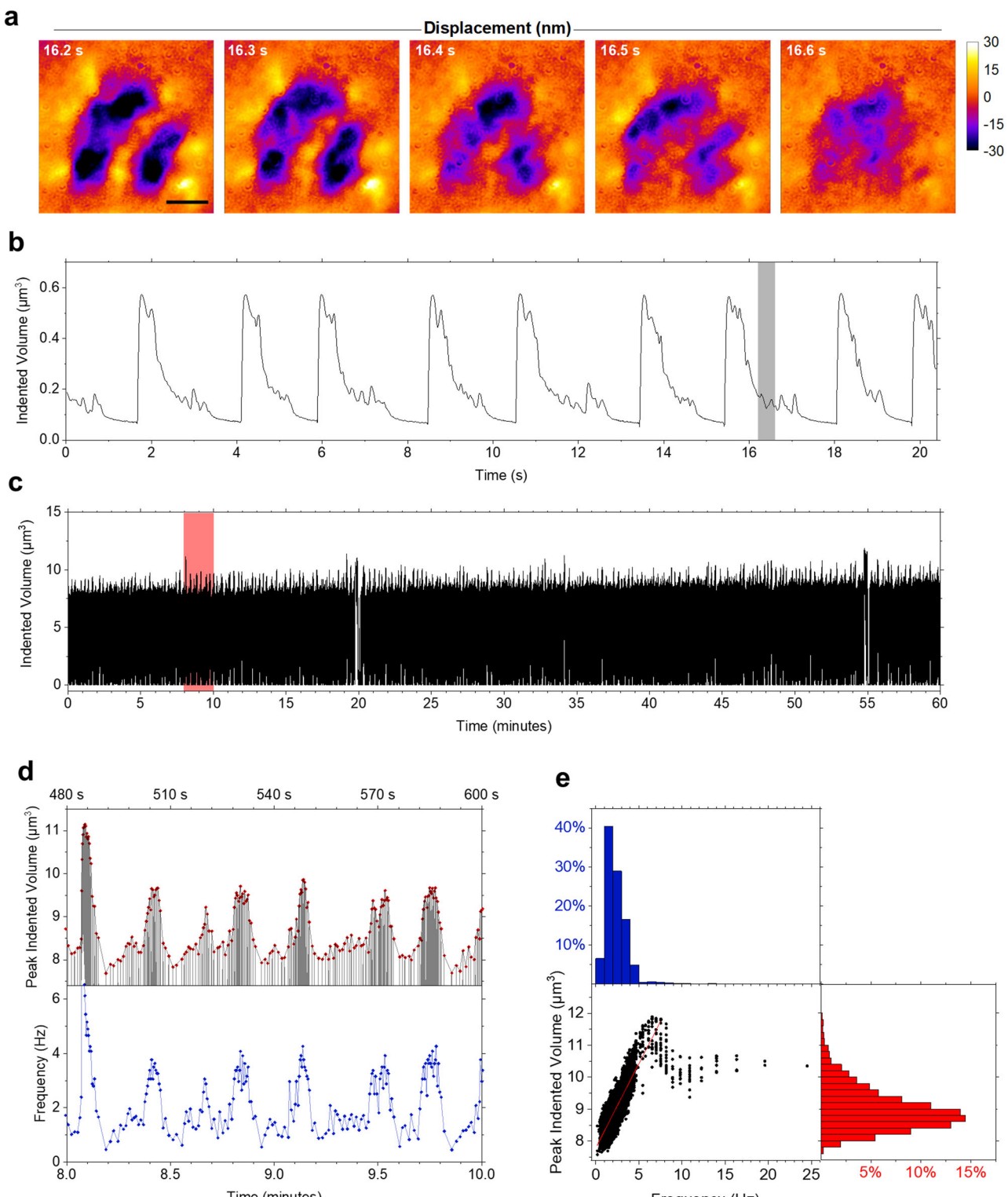

**Fig. 5 WARP measurements across different scales. a** Section from a 100 fps WARP measurement of cardiomyocytes with characteristic micro-contractions during the resting phase between the main contractions. Spatial Fourier-filtering was applied to filter out the broad, contraction-induced deformation patterns. **b** Temporal evolution of the indented volume for cells in **a** over nine contraction cycles. The section marked by the grey box corresponds to the period shown in **a**. **c** Evolution of indented volume in the central region of a cell taken from a 1 h long-term measurement. **d** Indented volume (grey line), and for each contraction cycle the peak indented volume (red dots) and contraction frequency (blue dots). Shown is the time interval marked by the red box in **c**. **e** Correlation analysis of peak indented volume and contraction frequency (centre panel, black dots). Red line indicates a linear fit to the data below 8 Hz ($R^2 = 0.76$). Histograms of peak indented volume (right panel, red bars) and contraction frequency (top panel, blue bars). Scale bar, 20 μm. Images are representative of two independent observations from one experiment.

provides the local stress. Statistical analysis of the data acquired from the images after conversion to displacement and stress was performed using Origin software (OriginLab).

**Real-time imaging.** The real-time analysis was performed using the ImageJ extension Micromanager, which allowed the Python analysis script to interface directly with the sCMOS camera using the MMCorePy wrapper. The Python package Tkinter was used to create a graphical user interface for the real-time measurements, whereas the library Matplotlib was used to create the real-time displacement map display. The background level to be subtracted from the sum image was identified by minimising abrupt and steep steps in thickness in the displacement map, which generally occur when a fringe is not correctly identified.

To achieve the processing speed necessary for real-time imaging, acquired images were split into separate regions and each region was analysed in parallel by a separate processor core, of which there were six in our case. This was achieved using the multiprocessing package in Python. For the NIH-3T3 fibroblast measurements, which involved scanning across the chip surface in real-time, $2 \times 2$ pixel binning was used to allow for the widest possible field of view and highest possible frame rate.

**Chip preparation.** The microcavity chips were manufactured using a procedure described earlier[17]. In brief, the bottom gold mirrors were prepared by depositing 0.5 nm of chromium, 10 nm of gold and 50 nm of silicon dioxide onto #5 cover glasses using electron beam evaporation. The elastomer (Nusil Gel8100) consisted of two components which were mixed in equal parts, before spin-coating the mixture at 3000 rpm for 60 s on top of the gold mirror to create an 8 μm thick layer. Afterwards the elastomer was crosslinked for an hour on a hotplate set to 125°C. The hydrophobic surface of the elastomer was oxidised using an oxygen plasma in a sputter chamber to create a hydrophilic layer, followed by the deposition of a 15 nm gold layer via thermal evaporation. Controlling the time and power of the oxidisation process prior to deposition of the final gold layer tuned the apparent stiffness of the microcavity surface, which was used to create chips with ≈5 kPa apparent stiffness (used for NIH-3T3 fibroblasts and for macrophages) and ≈6 kPa and ≈14 kPa apparent stiffness (used for cardiomyocytes). Stiffness measurements were performed by AFM (Nanosurf FlexAFM), following the protocol described in ref. [17] (see Supplementary Note 11 for an example). Before culturing cells on top of the microcavity, four silicone chambers (ibidi), each with a capacity of 400 μL, were attached to the surface.

**Cell culture.** NIH-3T3 fibroblasts were trypsinized from a cell culture flask, centrifuged, suspended in fresh medium (phenol red-free Dulbecco's Modified Eagle Medium (DMEM) supplemented with 10 vol% fetal bovine serum (FBS), 1 vol% penicillin/streptomycin solution and 1 vol% 100× concentration Glutamax) and seeded onto the microcavity chip. No additional coating was applied to the microcavity chips, as NIH-3T3 fibroblasts adhered well to the surface of the gold mirror. Cells were incubated overnight before performing the live measurements the following day.

To obtain primary macrophages peripheral blood was obtained from a normal healthy donor after ethical review (School of Medicine, University of St Andrews) and informed consent. Peripheral blood mononuclear cells were purified using a Ficoll gradient, and then adhered to plastic culture dishes at 37°C for 60 min in RPMI 1640 (Invitrogen) supplemented with 5 vol% FBS (Invitrogen). After extensive washing to remove lymphocytes, the adherent monocytes were cultured in 50 ng/ml granulocyte-macrophage colony-stimulating factor (Immunotools) to generate macrophages. Cells were detached from the culture flask by incubation with trypsin for 30 min under repeated gentle tapping, centrifuged and resuspended in fresh serum-free medium (99 vol% DMEM, 1 vol% penicillin–streptomycin) to encourage podosome activity. Cells were seeded onto microcavity chips and incubated overnight prior to WARP imaging.

To obtain primary cardiomyocytes, heart tissue was collected from postnatal Day 1–3 C57 mouse pups (mice were housed at 20–24°C and 45–65% relative humidity on a 12 h light-dark cycle) in ice-cold $Ca^{2+}/Mg^{2+}$-free Dulbecco's PBS, incubated in 10 U ml$^{-1}$ papain for 30 min at 37°C then triturated in culture medium (DMEM supplemented with 10 vol% FBS, 1 vol% non-essential amino acids, 1 vol% 100× concentration Glutamax and 1 vol% penicillin/streptomycin). The initial cell suspension was seeded on an uncoated culture dish for 2 h, to which fibroblasts adhered while cardiomyocytes predominantly remained in solution. The cardiomyocyte-enriched suspension was then pelleted by centrifugation at $200 \times g$ and resuspended in a fresh medium. Microcavity chips were coated with 5 μg ml$^{-1}$ fibronectin in 0.02% gelatine at 37 °C for 1 h before seeding cardiomyocytes. Cultures were incubated for 2–3 days before performing force measurements. For nifedipine challenge experiments, the nifedipine concentration was ≈500 nM. To wash out nifedipine, 200–250 μL was extracted from the chamber, before adding 300 μL of fresh, pre-warmed medium; this was repeated three times. The use of experimental animals was approved by the Animal Ethics Committee of the University of St Andrews. All animal procedures conformed to Directive 2010/63/EU of the European Parliament on the protection of animals used for scientific purposes and the United Kingdom Animals (Scientific Procedures) Act 1986.

## Data availability

The data sets supporting this publication can be accessed via the PURE repository of the University of St Andrews at https://doi.org/10.17630/9b706387-c972-4c87-8c48-f492192590b1.

## Code availability

The scripts written for the WARP analysis are available via the PURE repository of the University of St Andrews at https://doi.org/10.17630/9b706387-c972-4c87-8c48-f492192590b1.

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

## Acknowledgements

This research was financially supported by an EPSRC Programme Grant (EP/P030017/1), by the European Research Council under the European Union's Horizon 2020 Framework Programme (ERC StG ABLASE, 640012) and by the EPSRC Doctoral Training Partnership (EP/M508214/1, EP/L505079/1). M.C.G. acknowledges funding from the Alexander von Humboldt Stiftung (Humboldt-Professorship).

## Author contributions

A.T.M. developed the alternating-wavelength setup, associated analysis algorithm and software and performed the WARP measurements. N.M.K., A.T.M. and J.H.B. prepared microcavity chips, and N.M.K. and E.D. performed AFM measurements. A.T.M., N.M.K. and M.C.G. developed the study and interpreted the data. A.M. prepared cardiomyocyte cultures and contributed to the cardiomyocyte measurements. P.L. contributed to the development of the analysis software. J.M. performed initial analysis of single wavelength reflection data. S.J.P. prepared primary macrophages and contributed to interpretation of macrophage data. M.C.G. coordinated the work. A.T.M. and M.C.G. wrote the manuscript with input from all authors.

## Competing interests

The authors declare no competing interests.
