## [Peer Review File · Nature Communications]

Reviewers' comments:

Reviewer #1 (Remarks to the Author):

In this manuscript, an interference-based imaging technique was reported and their application in obtaining high speed force measurements are demonstrated. The system is based on a deformable mirror surface on which cells are cultured, and whose deformation create changes in interference pattern that can be back-calculated into force. Two wavelengths were used to resolve ambiguity in analysis, and a calibration curve is used to look up the displacement value. This configuration allows high-speed imaging of the displacement.

Overall this is an interesting approach for realtime imaging of cellular forces, that builds upon a previous study by the authors (Kronenberg et al., Nature Cell Bio 2017). However, in the current form my opinion the manuscript should be substantially revised before being suitable for Nature Communications.

Firstly, it is currently formatted as a short report with a very short abstract and few number of main figures, which makes it somewhat challenging to follow. A large part of the analysis of the performance of the technique are in the supplementary materials.

Second, while the technique was touted as a 'force'-imaging technique with picoNewton precision, in many figures, displacements in nm are shown instead. The seemingly interchangeable use of stress and displacement is confusing. The results should be converted into stress (Pa) or picoNewton since it is more readily biologically interpretable while displacement is more contingent upon the experimental system configuration. Also, if 'picoNewton precision' will be claimed in the title, perhaps a validation by AFM or other force-measurement techniques should be carried out. In the current version, it appears the authors just used the calibration carried over from their earlier work.

Thirdly, related to the previous point, the measurement of force exerted by podosome has also been reported by an alternate method, Protrusion Force Microscopy (Labernadie et al. Nature Comm. 2014) using an AFM-based approach. While the author cited this in their earlier ERISM work, this was somehow omitted in the reference of this manuscript. In that study, the podosome protrusion generates a displacement in the nm range which seems comparable to this study. However, the force magnitude in that study was calculated to be in the range of hundreds of nN per podosomes. Given the different representation/unit used in that study versus in this study, it is difficult to draw comparison. One important suggestion for this manuscript is to calculate the actual force exerted per podosome in Newton scale, so that this can be compared with such previous studies. The similarity and differences between these approaches would also merit some detailed discussion.

Fourth, measurements of force exertion by podosomes, focal adhesions, and cardiomyocytes have been done by various approaches previously. In the revision, it would be helpful to compare and discuss the measurements reported here with these previous studies. If possible, some effects on the force by various perturbation (e.g. such as in Labernadie study) could be demonstrated in real-time.

Reviewer #2 (Remarks to the Author):

Review of article titled "Real-time imaging of cellular forces with piconewton precision" by Meek et al.

The paper demonstrates a sped-up version of ERISM – the Gather group development for traction force imaging using a Fabry-Perot resonator. The improvement in speed is impressive though I do not find the particular demonstrations very convincing as to why I (or the community) will need this capability. Moreover, the amount of useful content in the SI that essentially proves the method validity should be moved the main text since it is otherwise asking the reader to take large leaps of faith in accepting the accuracy of the improved system. I have a number of major and

minor remarks to improve the paper, which might make it appropriate for Nature Communications.

Major Comments:

1. While the paper is about an instrumentation advance, there is almost zero space in the main text devoted to explaining the new concepts and methods that allowed for the more than 100-fold speed up for this device. The SI, specifically parts of Section 1 and 2 and Figure 13 need to be in the main text after figure 1 – which itself should be broken up into multiple figures. This part of the SI explains the method clearly and shows the validation for the method compared to the authors standard – their old method.
2. I find the demonstrations of pN sensitivity a bit misleading. The entire concept of force sensitivity is a bit odd in this case (and in the previous work) since this is intrinsically tied to the compliance of the substrate. The real power here is, as the authors state, that they measure axial distance with high precision using interferometry – one already demonstrated quite a while back in Fourier Domain OCM. Whether the substrate is 1 kPa or 100 GPa is what determines the force resolution. The authors need to revise their presentation to stick with their primary achievement, which is measuring small axial displacements very fast.
3. The validation experiments are almost nonexistent. I understand the ERISM paper is already published, but some validation needs to be shown here. Can the authors construct a resonator in a wedge shape and show the height profile? I guess this is sort of done in Figure 1 f, but the explanation is very poor and no schematic even shows the ground truth. There is no comparison of Figure 1 g with the ground truth. How can the authors know if their method actually works?
4. I find the transition from the cotangent image profiles in Figure 1e to a relative thickness in Figure 1 f very difficult to follow. I read the math in the SI, and I understand the basic idea, but I do not understand how one cotangent curve is distinguished from another across different lateral positions? They all look the same to me except that their width changes because of the changing thickness of the resonator. The authors should spend some time, and maybe a figure, to explain how the cotangent image is converted in the relative height and how this is used across an entire image at different lateral positions. Finally, the authors need to also list some of the ambiguities with this method.
5. I believe a long-term demonstration of force mapping on a crawling neutrophil or similar motile cell is necessary to show the versatility of this technique for general cell biophysics studies. I believe they can compare their results from TFM with existing literature to determine the accuracy and show the improved ability to measure long-term, noninvasively, and with high speed (and accuracy). I do like the demonstration with cardiomyocytes, but for an instrument paper, another demonstration is really required where one can make quantitative comparisons. Why did the authors choose the cardiomyocyte example?

Minor comments:

6. The authors fail to cite any of the super resolution techniques applied to molecular tension sensors in the field. Some of those references can be found here:
<https://www.sciencedirect.com/science/article/pii/S2468451117300776>.
7. The order of panels in Figure 1 is unideal. Either choose left to right for lettering or top to bottom.
8. The English language needs to be checked for errors, which there are few. I think a native speaker can do this relatively quickly.

Reviewer #1 (Remarks to the Author):

In this manuscript, an interference-based imaging technique was reported and their application in obtaining high speed force measurements are demonstrated. The system is based on a deformable mirror surface on which cells are cultured, and whose deformation create changes in interference pattern that can be back-calculated into force. Two wavelengths were used to resolve ambiguity in analysis, and a calibration curve is used to look up the displacement value. This configuration allows high-speed imaging of the displacement.

Overall this is an interesting approach for realtime imaging of cellular forces, that builds upon a previous study by the authors (Kronenberg et al., Nature Cell Bio 2017). However, in the current form my opinion the manuscript should be substantially revised before being suitable for Nature Communications.

We thank the reviewer for their generally positive appraisal and have thoroughly revised our work as described below.

Firstly, it is currently formatted as a short report with a very short abstract and few number of main figures, which makes it somewhat challenging to follow. A large part of the analysis of the performance of the technique are in the supplementary materials.

The very condensed formatting of the original manuscript is due to an automated transfer from a Brief Communication in Nature Methods to Nature Communications. In our revised manuscript, we have expanded the abstract, and moved several sections and figures from the Supplementary Information to the main body of the paper; these discuss in more detail the algorithm, analysis procedure and accuracy of our method. We have also split up the two large main figures into separate figures. Combined we believe that this makes our manuscript much more accessible.

Second, while the technique was touted as a 'force'-imaging technique with picoNewton precision, in many figures, displacements in nm are shown instead. The seemingly interchangeable use of stress and displacement is confusing. The results should be converted into stress (Pa) or picoNewton since it is more readily biologically interpretable while displacement is more contingent upon the experimental system configuration. Also, if 'picoNewton precision' will be claimed in the title, perhaps a validation by AFM or other force-measurement techniques should be carried out. In the current version, it appears the authors just used the calibration carried over from their earlier work.

We appreciate this concern and have addressed this point in several ways as described in the following.

While the imaging system and analysis routine described in our manuscript are novel, the microcavities used are of the same design as the ones used in a prior study [Ref. 17 in our manuscript]. As in our previous study, stiffness calibrations for the used microcavities were performed using an AFM as suggested by the referee. Force-distance curves were recorded with a spherical indenter mounted to the AFM head and the results were fitted with a corrected Hertz model. Given this was executed as previously described, we did not discuss the mechanical calibration of the microcavities in detail again. However, we appreciate that the reader might want to see calibration for the batch of microcavities

used here. The revised manuscript therefore shows representative AFM measurements in the Supplementary Information (Supplementary Note 10).

Displacement (and the indented volume used in some parts of our analysis) are directly correlated with the stress and the forces exerted on the cavity surface by the cells. The conversion from displacement to stress requires application of a mechanical finite element model (FEM), which is computationally demanding and currently cannot be performed in real-time. Therefore, displacement and indented volume are used in place of force and stress in instances where the real-time capability of our technique is important, or where the large number of frames would preclude conversion to stress / force on useful time scales. Improving the computation times required for FEM was not the focus of the present work, but we expect that significant improvements could be made here. [One strategy may be to use machine learning to train an A.I. with a set of displacement maps and the corresponding stress maps obtained by FEM and then have the AI system estimate stress from displacement.] For now, as the stiffness of the elastic microcavity used in our study was consistent between measurements, we used displacement and indented volume as a quantitative measure of the level of stress exerted by cells in most cases.

Where computation time is not critical, displacement maps can be converted to stress maps and exerted forces can be extracted by integrating over the relevant regions of the stress map. In our original manuscript, we have briefly demonstrated this for the example of macrophage podosomes. In our revised manuscript, we have extended the analysis and discussion of stress and force for these cells, now studying the temporal evolution of the force exerted by podosomes in detail and investigating the correlation between podosome force and the area over which podosomes indent (Fig. 3d-g and corresponding description).

In addition, to avoid confusion and acknowledging that the piconewton force sensitivity is not the most crucial aspect of our present work, we have reworded our claims in the abstract and have omitted the claim of picoNewton resolution in the title of our paper, which now reads “Real-time imaging of cellular forces using optical interference”.

Thirdly, related to the previous point, the measurement of force exerted by podosome has also been reported by an alternate method, Protrusion Force Microscopy (Labernadie et al. Nature Comm. 2014) using an AFM-based approach. While the author cited this in their earlier ERISM work, this was somehow omitted in the reference of this manuscript. In that study, the podosome protrusion generates a displacement in the nm range which seems comparable to this study. However, the force magnitude in that study was calculated to be in the range of hundreds of nN per podosomes. Given the different representation/unit used in that study versus in this study, it is difficult to draw comparison. One important suggestion for this manuscript is to calculate the actual force exerted per podosome in Newton scale, so that this can be compared with such previous studies. The similarity and differences between these approaches would also merit some detailed discussion.

We thank the reviewer for pointing out this omission which was triggered by the tight limit on the number of allowed references at Nature Methods. In our revised manuscript we now cite the Labernadie et al. paper (Ref. 22 in revised manuscript). Deeper understanding of podosome mechanobiology was not the focus of our present study, but our revised manuscript includes a concise discussion of the similarities and differences between the work by Labernadie et al. and our work.

We have also now calculated the force exerted per podosome from our data. From this, we observed a correlation between the podosome force and area over which the podosome indents. Additionally, we observed fast fluctuations in podosomal forces on timescales < 5 s.

Our results show similarities to the observations described by Labernadie et al., including the correlations they observed between podosome force, height and area, and the fluctuations they measured for the podosome height on short timescales.

As pointed out by the referee, the absolute value of the displacements generated by podosomes are comparable in both studies. However, the absolute value of the force exerted differs substantially. We attribute this difference predominantly to a drastic difference in the effective stiffness of the used substrate between the two studies, i.e. podosomes appear to indent by comparable depth, independent of the local substrate stiffness.

Fourth, measurements of force exertion by podosomes, focal adhesions, and cardiomyocytes have been done by various approaches previously. In the revision, it would be helpful to compare and discuss the measurements reported here with these previous studies. If possible, some effects on the force by various perturbation (e.g. such as in Labernadie study) could be demonstrated in real-time.

In our revised manuscript, we now discuss and compare our cardiomyocyte results to the work of prior groups (which used elastic micropillars and TFM), in particular with respect to the effects of substrate stiffness and nifedipine challenge on cardiomyocyte behavior. Our results are consistent with the prior studies, but in several cases provide new information that would be difficult or impossible to obtain with the methods used in prior studies. The nifedipine challenge, which is discussed in detail in our revised manuscript, represents a real-time perturbation of cells and indeed we observe profound effects on the cellular forces.

Furthermore, as described in the response to the previous point, we have included a comparison between our macrophage podosome datasets and the work by Labernadie et al.

Reviewer #2 (Remarks to the Author):

Review of article titled "Real-time imaging of cellular forces with piconewton precision" by Meek et al. The paper demonstrates a sped-up version of ERISM – the Gather group development for traction force imaging using a Fabry-Perot resonator. The improvement in speed is impressive though I do not find the particular demonstrations very convincing as to why I (or the community) will need this capability. Moreover, the amount of useful content in the SI that essentially proves the method validity should be moved the main text since it is otherwise asking the reader to take large leaps of faith in accepting the accuracy of the improved system. I have a number of major and minor remarks to improve the paper, which might make it appropriate for Nature Communications.

We thank the referee for their constructive review of our work and for their useful suggestions. We have addressed these as described in detail below.

Major Comments:

1. While the paper is about an instrumentation advance, there is almost zero space in the main text devoted to explaining the new concepts and methods that allowed for the more than 100-fold speed up for this device. The SI, specifically parts of Section 1 and 2 and Figure 13 need to be in the main text after figure 1 – which itself should be broken up into multiple figures. This part of the SI explains the method clearly and shows the validation for the method compared to the authors standard – their old method.

The very condensed description in the original manuscript originates from the fact that the manuscript was originally prepared as a Brief Communication in Nature Methods and then transferred to Nature Communications. In our revised manuscript, we have broken up Fig. 1 (now forming Figs 1 and 3) and have moved material from the former Sections 1 and 2 of the Supplementary Information to the main text. The key information from the figures in these sections and from the former Supplementary Fig. 13 is provided in Fig. 2 of our revised manuscript. We trust that this substantial rearrangement allows the reader to understand and assess the general concept and the basic implementation of our algorithm from the main text, with the revised Supplementary Information providing additional background information that will be of interest for a more specialized audience.

2. I find the demonstrations of pN sensitivity a bit misleading. The entire concept of force sensitivity is a bit odd in this case (and in the previous work) since this is intrinsically tied to the compliance of the substrate. The real power here is, as the authors state, that they measure axial distance with high precision using interferometry – one already demonstrated quite a while back in Fourier Domain OCM. Whether the substrate is 1 kPa or 100 GPa is what determines the force resolution. The authors need to revise their presentation to stick with their primary achievement, which is measuring small axial displacements very fast.

The force sensitivity of our system is given by both the compliance of the substrate and by the precision of the axial distance measurement. We agree that the key novelty here is the ability to perform local distance measurements with high precision and high speed by using a wide-field interferometry method based on illumination with alternating wavelengths. While there are similarities to Fourier Domain OCM, the two are distinctly different, with the latter generally not being wide-field and requiring the point-scanning acquisition of a very large number of spectra rather than just two wide-field images.

We would maintain that because the application of our method is in mechanobiology and because cells often react to the compliance of their substrate, compliance is not a parameter that can be varied arbitrarily to achieve any desired force resolution. To mimic the compliance of biological tissue and to resolve relevant mechanical behaviour of cells, a substrate compliance in the kPa range and (at least in many cases) a force sensitivity in the deep sub nN range are necessary. Achieving this requires the level of precision afforded by an interferometric measurement. Furthermore, making an optical device that mimics the rather low compliance of biological tissue is challenging from a device fabrication perspective.

Upon further reflection, we agree with the referee that despite the above points, the original manuscript put the force resolution too much into the foreground. To address this, we have reworded parts of the abstract and have changed the title of our paper to “Real-time imaging of cellular forces using optical interference”, thus removing the pN claim there. We then discuss the absolute values of

the measured forces in more detail where they are relevant to the biology, in particular in the section on podosomes (Fig. 3).

3. The validation experiments are almost nonexistent. I understand the ERISM paper is already published, but some validation needs to be shown here. Can the authors construct a resonator in a wedge shape and show the height profile? I guess this is sort of done in Figure 1 f, but the explanation is very poor and no schematic even shows the ground truth. There is no comparison of Figure 1 g with the ground truth. How can the authors know if their method actually works?

The calibration of our method is performed by comparing an ERISM thickness measurement across a region of the microcavity, where displacement varies gradually, against the numerical values in the division image (i.e. the image obtained by dividing the sum and difference of the images obtained at two different wavelengths, formerly referred to as cotangent image). This is generally performed at the edge of the measurement well, where the weight of the silicone chamber used for containing cells causes the surface to slope downwards thus forming a wedge. Because the data shown in Fig. 1f,g is used for calibration, we did not use it for validation as this would seem to make a circular argument.

Instead, to validate our method against a ground truth, Fig. 2 of the revised manuscript compares the displacement exerted by a static cell as measured with our new method against the displacement measured by ERISM. Using ERISM as ground truth here is justified because it uses basic interferometry to extract the absolute thickness at each position across the microcavity (i.e., it truly uses the wavelength of light as the ruler) and because in our original publication on ERISM (Ref. 17), we also validated ERISM against AFM.

4. I find the transition from the cotangent image profiles in Figure 1e to a relative thickness in Figure 1 f very difficult to follow. I read the math in the SI, and I understand the basic idea, but I do not understand how one cotangent curve is distinguished from another across different lateral positions? They all look the same to me except that their width changes because of the changing thickness of the resonator. The authors should spend some time, and maybe a figure, to explain how the cotangent image is converted in the relative height and how this is used across an entire image at different lateral positions. Finally, the authors need to also list some of the ambiguities with this method.

It was useful to learn that this part was not clear from our original manuscript and we would like to clarify as follows:

We do not distinguish the different cotangent shaped curves, and their shape indeed remains nearly unchanged across different lateral positions, i.e. for the different modes of the cavity. However, whenever we move from one cotangent curve to the next, i.e. whenever we encounter a sharp fringe with a transition from -ve to +ve in the cotangent image (referred to as 'division image' in the revised manuscript), we accumulate a change in cavity thickness of approximately $\lambda/(4n)$ (approximately 110nm). To create the displacement map, our algorithm runs across the division image laterally and adds/subtracts a displacement of $\lambda/(4n)$ at each fringe (addition at -ve to +ve fringes and subtraction at +ve to -ve transitions). In the manuscript, we refer to this process as automated fringe counting.

In order to compute the accurate displacement at positions in between fringes, we use the calibration described in response to the referee's previous point. Here, we can exploit the fact that the shape of the

cotangent shaped curve remains unchanged for different modes of the cavity (at least across practically relevant ranges of displacement; see Supplementary Fig. 8 in the revised manuscript).

In the revised manuscript, the conversion algorithm is discussed in much greater detail in the main text. In particular, the main text now explains the algorithm through a worked example, with the various steps applied for a single cell measurement (Fig. 2a-g). At the end of this, a comparison is made to an ERISM measurement at the same position (Fig2h,i). The main text now explicitly points out that potential ambiguities are discussed in Supplementary Note 2. The discussion of ambiguities itself is somewhat lengthy due to the complexity of this matter. As it ultimately shows that ambiguities are unlikely to have a bearing for the used cavity thicknesses and the displacements expected from cells, we have refrained from moving this information into the main text.

5. I believe a long-term demonstration of force mapping on a crawling neutrophil or similar motile cell is necessary to show the versatility of this technique for general cell biophysics studies. I believe they can compare their results from TFM with existing literature to determine the accuracy and show the improved ability to measure long-term, noninvasively, and with high speed (and accuracy). I do like the demonstration with cardiomyocytes, but for an instrument paper, another demonstration is really required where one can make quantitative comparisons. Why did the authors choose the cardiomyocyte example?

The principal novelty of our method over existing techniques, including the ERISM method previously published by us, is the ability to acquire maps of mechanical activity at high speed, while also achieving high accuracy and avoiding light-induced cell damage. Cardiomyocytes were chosen to test our method because (1) they exhibit a highly dynamic behaviour and thus exert forces that vary on a sub-second timescale, (2) it is important to have the ability to monitor changes in the dynamic behaviour of cardiomyocytes over extended time scales (minutes, hours), e.g. to test drug response, and (3) we found that cardiomyocytes exert small yet also highly dynamic forces in between the main contraction cycles which are likely of interest in the context of SPOC waves.

Another example where the high acquisition speed and non-invasiveness of our method are of great benefit is for the real-time acquisition of deformation maps, e.g. to identify cells of interest in a culture dish or to perform high throughput testing. Experiments demonstrating this ability are shown in Fig. 3a-c and more clearly in Supplementary Videos 1 and 2. To our knowledge real-time acquisition of mechanical activity maps has not been demonstrated with other force mapping methods so far.

Studying the force pattern of motile cells (e.g. neutrophils) is of great importance and has been pursued extensively over at least the last two decades, using TFM and other methods. However, in our view the ability to acquire at high speed is unlikely to add substantial insights here. In the past, we have used ERISM – the ‘slower analogue’ of the method introduced here – to continuously monitor cellular migration over long periods of time (up to a week). Repeating these demonstrations with our new method seems redundant at the present stage as we have shown that for slow or static processes the information obtained with the two methods is equivalent and both methods are identical in terms of non-invasiveness.

In our revised manuscript, we compare our results to the work of others and – where conditions are comparable – obtain consistent results. Specifically, we discuss earlier results obtained with elastic

micropillars and TFM that also investigated the effects of substrate stiffness and nifedipine challenge on cardiomyocyte behaviour. Furthermore, we have included a comparison between our macrophage podosome datasets and the work of Labernadie et al.

Minor comments:

6. The authors fail to cite any of the super resolution techniques applied to molecular tension sensors in the field. Some of those references can be found here:

<https://www.sciencedirect.com/science/article/pii/S2468451117300776>.

We originally cited one example of a FRET based molecular tension sensor which measures intracellular forces in real time. We have now expanded on this a bit more and included the above review and a recent paper that makes use of super resolution techniques (Refs. 15, 16).

7. The order of panels in Figure 1 is unideal. Either choose left to right for lettering or top to bottom.

In the revised manuscript, we have split Figure 1 into two separate figures to better explain the method and to make the figures easier to follow. We believe that this also addresses the issue of the order of the panels.

8. The English language needs to be checked for errors, which there are few. I think a native speaker can do this relatively quickly.

We have carefully revised our manuscript for language errors and had it proofread by another native speaker. We trust that the quality of English is now acceptable.

REVIEWER COMMENTS

Reviewer #1 (Remarks to the Author):

I have reviewed the revised manuscript, and while the authors have undertaken a significant re-writing and re-arrangement to make it more accessible to readers, it would seem that the key scientific concerns that I raised on the original version has not been addressed sufficiently, and thus unfortunately at this point I am still not able to support this manuscript for Nature Communications.

The crux of the matter is that I am still not able to assess how reliable this WARP method is, given the limited information provided. Importantly, the authors describe only one scenario in which the force magnitude is calculated (the podosome). This force magnitude is 4 orders of magnitude!!! lower than the measurements by a different technique (in Fig. 1 Labernadie et al., Nat. Comm. 2014) in the same cell type (i.e, ~20 pN vs. ~100 nN). I was hoping that in the revision the authors will address this directly and look into other structures where the force magnitudes are known (e.g. focal adhesions), which should already be recorded in the raw data for cardiomyocytes and NIH3T3. However, as the authors insist on not calculating the force for these other datasets despite the serious discrepancy above, my major concern about the reliability of this WARP technique is still not addressed.

Detailed comments:

1. As stated since the review of the original version, in my opinion the calculation of force is obviously an essential component of any force-measurement technique. For this manuscript, the importance of force calculation is clear given how the authors prominently mention force in the title and every sentence of the abstract. Given the lack of such force calculation, my opinion is that in the current form, the technique is interesting as a high-speed label-free displacement measurement method, but its suitability as a force-measurement approach is still lacking and not adequately demonstrated.
2. How accurate are the force measurement results? The podosome results are not even close to the expected range and this raise more questions than answers (more below). The burden of proof for the validity of the technique should be on the authors. However, the authors simply leave out the FEM analysis to obtain the force to the readers.
3. The calculation of force from displacement is not a trivial challenge, and in techniques such as traction force microscopy, extensive theoretical analyses have been investigated to establish the validity of the techniques (e.g. Sabass...Schwarz., Biophys J 2007; Han...Danuser, Nature Methods 2015). Off-line analysis is a commonly accepted practice as there are well-defined computational pipelines. However, for this manuscript the authors did not document this aspect in adequate details for independent reproducibility (reliability, computation cost, workflow, parameters, etc.). Indeed, the WARP technique is reminiscent of the classic silicone wrinkle study (Harris... Stopak, Science 1980), which pioneered the study of cellular force transmission based on observation of wrinkle pattern. However, the challenges in calculating force from the wrinkle pattern severely limits their adoption, and in modern-day research it is superseded by traction force microscopy or molecular tension gauge approaches which can provide force measurements in a more tractable manner. If the force calculation challenges for WARP is not addressed, my concern is that this technique will remain highly specialized and of limited appeal to the community.
4. As above, the Podosome force measurements by the authors differ from the previous study by >4 orders of magnitude, despite most other podosome properties being comparable. Given that this is the only instance where the actual force magnitude is calculated, in my opinion this seriously undermine the reliability of WARP to obtain accurate force magnitude. Unfortunately, in the revision this serious discrepancy was largely brushed aside without rigorous evaluation: "While the depth of podosome protrusion was comparable in both studies, podosomal forces differed significantly, presumably because cells were cultured on substrates with different compliance"

Based on a large body of work showing that force magnitude at a single-molecule level is already on the 5-10 piconewton range (e.g. by FRET tension sensors), the 20 pN magnitude for the entire podosome seems anomalously low, especially since podosomes are large bundles of hundreds of

not more actin filaments, and whose biological functions include force exertion to invade the underlying matrix. While there are certainly differences in the substrate stiffness between this manuscript and the Labernadie study, if both measurements are valid this would imply a podosome force dynamic range of over 4 orders of magnitude. This would seem to be quite an unrealistically broad range for a biological structure but if verified could be very interesting biologically.

5. In the literature, focal adhesions have been shown to apply nN range of force which is closer to the Labernadie measurement of ~ 100 pN for podosomes. A key question would then be whether the NIH3T3 or cardiomyocytes studied in this manuscript exert similar level of forces through their focal adhesions. However, in the figures, only displacement are shown, and there are no details analysis in the focal adhesion regions. As the authors still insist on not providing the force calculation, this deprives the readers of additional datapoints to assess the validity of the technique.

6. One of the benefits of WARP claimed by the author is that it is "label-free". However, in practice fluorescent imaging is a very powerful capability for marking different cell types or indicating different biological states. A question is thus whether the WARP cavity which is coated by a gold layer is still compatible with fluorescence imaging or not.

Reviewer #2 (Remarks to the Author):

The revised paper is much better in conveying the actual physics behind the method, and I appreciate the authors thoughtful responses to the comments. However, I believe that the methodology should be further explained before publication.

1. I feel that the method is the going from the division images to nm displacements is crux of the paper. The demonstrations on cells are fine, and to be honest, no one can really argue with them because part of it has never been measured before. Nevertheless, I think it is imperative that the authors bring in Supp. Note 3 - how to actually get a displacement map into the paper. This is the most important part of the work. Without the mapping between their cotangent to distance, the authors have very little to discuss.

2. In addition to the comment above, I still feel that a ground truth quantitative comparison is necessary to validate the method.

Reviewer #1 (Remarks to the Author):

I have reviewed the revised manuscript, and while the authors have undertaken a significant re-writing and re-arrangement to make it more accessible to readers, it would seem that the key scientific concerns that I raised on the original version has not been addressed sufficiently, and thus unfortunately at this point I am still not able to support this manuscript for Nature Communications.

The crux of the matter is that I am still not able to assess how reliable this WARP method is, given the limited information provided. Importantly, the authors describe only one scenario in which the force magnitude is calculated (the podosome). This force magnitude is 4 orders of magnitude!!! lower than the measurements by a different technique (in Fig. 1 Labernadie et al., Nat. Comm. 2014) in the same cell type (i.e, ~20 pN vs. ~100 nN). I was hoping that in the revision the authors will address this directly and look into other structures where the force magnitudes are known (e.g. focal adhesions), which should already be recorded in the raw data for cardiomyocytes and NIH3T3. However, as the authors insist on not calculating the force for these other datasets despite the serious discrepancy above, my major concern about the reliability of this WARP technique is still not addressed.

We thank the referee for taking time to study our revised manuscript. We would like to make two remarks on the context of our work before addressing each concern in detail below.

- 1) As stated in our manuscript, we have previously introduced an interferometric force imaging method (Elastic Resonator Interference Stress Microscopy, ERISM) which uses the same type of elastic microcavity as the substrate for cells as WARP. ERISM obtains displacement measurements by acquiring at least 50 images at different illumination wavelengths. The crucial advance of the WARP method over ERISM is to reduce this requirement to just two images. Combined with a number of other optimizations, this has allowed us to increase the acquisition rate from 0.4 fps to 100 fps. In Fig 2h and i, we show that measurements with ERISM and WARP give the same displacement maps (within the error of the measurement).

Once a displacement map has been obtained with WARP, the further data analysis is identical to what has already been reported in the peer-reviewed literature for ERISM, see e.g. [Kronenberg, Nature Cell Biology 2017]. In particular, our earlier paper already finds ~20 pN forces for podosomes in macrophages. Furthermore, we recently published a study on invadopodia, which are closely related to podosomes, where we find mean forces of 5-10 pN [Dalaka, Science Advances 2020]. A further study, currently under review, finds that podosomes in epithelia cells exert forces of 20-40 pN [Diaz, bioRxiv <https://doi.org/10.1101/780874>].

- 2) In 2014, Labernadie et al. proposed a different method to study podosome protrusions (protrusion force microscopy, PFM). In this method, macrophages are cultured on one side of a thin membrane and the topography of this membrane is then recorded by AFM from the backside of the membrane. To our knowledge, this method has not yet been validated by other groups or against other methods. There are two further papers by the same group on the method [Bouissou, ACS Nano 2017; JOVE 2018]. Interestingly, in the 2017 paper, podosome forces reduce to 5-10 nN (versus 100 nN in the original publication).

Depending on which data one compares exactly, there is indeed a 2 to 4 orders of magnitude difference in podosome force between the two methods. However, the

stiffness of the substrates on which the cells are grown also differs by many orders of magnitude (ERISM/WARP ca. 5 kPa, consistent with the stiffness of many types of tissue; protrusion force microscopy ca. 2 GPa in a 60 nm thick membrane which we estimate corresponds to a bulk material with about 4-15 MPa stiffness for a podosome-sized indenter). As we point out in our manuscript, the vertical depth of the protrusions achieved by podosomes is very similar on the two different substrates. A stark difference in the force required to achieve these protrusions is therefore expected, and it is thus likely that the results by both groups are correct and not in contradiction. Labernadie et al. indeed report that podosomes are “mechanosensitive” and adjust the magnitude of force they exert according to the stiffness of the substrate.

In order to prevent confusion on this issue, we have extended the discussion of the podosome forces measured by both groups in the main part of the manuscript (lines 189-198) and added further validation of the WARP force calculation via AFM (Supplementary Note 11) and by a consistency check against a simply Hertz model (Supplementary Note 5).

Detailed comments:

1. As stated since the review of the original version, in my opinion the calculation of force is obviously an essential component of any force-measurement technique. For this manuscript, the importance of force calculation is clear given how the authors prominently mention force in the title and every sentence of the abstract. Given the lack of such force calculation, my opinion is that in the current form, the technique is interesting as a high-speed label-free displacement measurement method, but its suitability as a force-measurement approach is still lacking and not adequately demonstrated.

The key priority of the present work is the high speed of acquisition. To reflect this, we have removed or toned-down statements on force sensitivity during our previous revision.

As described above, we use a peer-reviewed method to convert displacement into force in our manuscript. To further address the reviewers concern about the magnitude of the reported force, we have now furthermore performed additional validation measurements, as described under the next point.

2. How accurate are the force measurement results? The podosome results are not even close to the expected range and this raise more questions than answers (more below). The burden of proof for the validity of the technique should be on the authors. However, the authors simply leave out the FEM analysis to obtain the force to the readers.

To address the accuracy of the FEM analysis of forces, we have now used atomic force microscopy (AFM) to show that the force as computed by FEM matches the force applied by an AFM cantilever (Supplementary Note 11). The results match within 10%, which further corroborates our podosome results. Additionally, we also show that applying a nN force to the microcavity chip by AFM, leads to an indentation of hundreds of nanometres (new Supplementary Fig. 18 a,b), substantially more than the indentations observed for podosomes, which are at most tens of nanometres (also see new Fig. 3d). (Using FEM to back-calculate the applied force from the displacement map results in the correct value.)

To provide a further check, we have also compared to the results from our FEM analysis with a simple model of a cylindrical Hertzian indenter (Supplementary Note 5).

We would also like to draw the referee's attention to the Methods section of our manuscript where we state that the FEM analysis is performed with the same procedure that is described in our earlier work. To avoid any confusion, we have now also amended the relevant statement in the main manuscript (lines 38-40):

"Once the local cavity displacement is known, finite element modelling (FEM) can be applied¹⁷ to compute a map of the mechanical stress that cells exert on the microcavity chip, in the same manner as we have previously described for ERISM".

3. The calculation of force from displacement is not a trivial challenge, and in techniques such as traction force microscopy, extensive theoretical analyses have been investigated to establish the validity of the techniques (e.g. Sabass...Schwarz., Biophys J 2007; Han...Danuser, Nature Methods 2015). Off-line analysis is a commonly accepted practice as there are well-defined computational pipelines.

However, for this manuscript the authors did not document this aspect in adequate details for independent reproducibility (reliability, computation cost, workflow, parameters, etc.). Indeed, the WARP technique is reminiscent of the classic silicone wrinkle study (Harris... Stopak, Science 1980), which pioneered the study of cellular force transmission based on observation of wrinkle pattern. However, the challenges in calculating force from the wrinkle pattern severely limits their adoption, and in modern-day research it is superseded by traction force microscopy or molecular tension gauge approaches which can provide force measurements in a more tractable manner. If the force calculation challenges for WARP is not addressed, my concern is that this technique will remain highly specialized and of limited appeal to the community.

We do not understand on which basis the referee likens the WARP method to the famous silicone wrinkle study by Harris et al. In fact, we find this comparison entirely unsubstantiated, as there is no analysis of wrinkles in our work, but displacements are determined through interferometric measurements and forces obtained using a FEM routine that is conceptually similar to other current work.

As stated above, the FEM force calculation used is discussed in detail in [Kronenberg, Nature Cell Biology 2017] (reference 17 in our manuscript) where we also provide the associated code. This description was not repeated here as the key technical innovation involved in the WARP study is the 100-fps displacement imaging rate, real-time imaging, and associated algorithm.

Indeed, calculation of force from displacement is associated with computational cost and off-line analysis is commonly accepted for this step. The more urgent shortcoming in our view is that there was up to now no method that allows real-time acquisition and visualization of displacement maps, in particular not for small forces and vertical forces. This is what our technique brings to the field. This point is openly discussed in our manuscript (lines 170-173):

"In addition, displacement maps were converted into stress maps using FEM (Fig 3.d; FEM is too computationally expensive to perform in real-time, but the resultant stress maps visually resemble the displacement maps, so a real-time assessment of stress can be made from the latter)."

4. As above, the Podosome force measurements by the authors differ from the previous study by >4 orders of magnitude, despite most other podosome properties being comparable. Given that this is the only instance where the actual force magnitude is calculated, in my opinion this seriously undermines the reliability of WARP to obtain accurate force magnitude. Unfortunately, in the revision this serious discrepancy was largely brushed aside without rigorous evaluation: “While the depth of podosome protrusion was comparable in both studies, podosomal forces differed significantly, presumably because cells were cultured on substrates with different compliance”

Based on a large body of work showing that force magnitude at a single-molecule level is already on the 5-10 piconewton range (e.g. by FRET tension sensors), the 20 pN magnitude for the entire podosome seems anomalously low, especially since podosomes are large bundles of hundreds if not more actin filaments, and whose biological functions include force exertion to invade the underlying matrix. While there are certainly differences in the substrate stiffness between this manuscript and the Labernadie study, if both measurements are valid this would imply a podosome force dynamic range of over 4 orders of magnitude. This would seem to be quite an unrealistically broad range for a biological structure but if verified could be very interesting biologically.

Only one other group has reported podosome forces so far, using a new method that has not been independently validated. (See discussion at the start of this response letter.) As discussed in the reply to comment (2), the AFM data shown in Supplementary Note 11 and the comparison to a simple Hertz model in Supplementary Note 5 both corroborate that our measurement of pN podosomal forces is correct under the conditions used in our experiment (in particular using a much softer substrate).

We have now further expanded our discussion of the Labernadie study and how it compares to our results, now stating (lines 189-198):

“Using AFM on the backside of thin membranes to which macrophages were adhered, Labernadie et al. also observed fluctuations in podosome protrusion on similar timescales²². While the depth of podosome protrusion was comparable in both studies, podosomal forces differed significantly, with the other work finding forces in the nN range. We attribute this difference to the fact that cells were cultured on substrates that differed in compliance by many orders of magnitude (WARP microcavity chips, ≈ 5 kPa; protrusion force microscopy, ≈ 2 GPa in a 60 nm thick membrane). As the vertical depth of the protrusions achieved by podosomes is very similar on the two different substrates, a stark difference in the force required to achieve these protrusions is expected, and the results reported by Labernadie et al. are thus not in contradiction to our findings.”

We agree that it would be interesting to study the ability of podosomes to adapt to different conditions in further detail. However, this is beyond the scope of the present study, which introduces a new method to rapidly and accurately record cell-induced displacements of substrates with physiological stiffness.

5. In the literature, focal adhesions have been shown to apply nN range of force which is closer to the Labernadie measurement of ~ 100 pN for podosomes. A key question would then be whether the NIH3T3 or cardiomyocytes studied in this manuscript exert similar level of forces through their focal adhesions. However, in the figures, only displacement are shown, and there are no details analysis in the focal adhesion regions. As the authors still

insist on not providing the force calculation, this deprives the readers of additional datapoints to assess the validity of the technique.

We assume what the referee intended to say here is “focal adhesions have been shown to apply nN range of force which is closer to the Labernadie measurement of ~100 nN (not 100 pN) for podosomes”.

The forces exerted by focal adhesions are predominantly horizontal to the substrate, with only a minor vertical component. By contrast, podosomes exert vertical forces, i.e. forces directed perpendicular with respect to the plane of the substrate. Like the Protrusion Force Microscope introduced by Labernadie et al, WARP is particularly well suited to quantify vertical forces because the optical microcavity is sensitive to vertical displacements.

In our previous work on ERISM, we have developed a modality to extract horizontal forces from our microcavity measurement and have compared our results for focal adhesions to the published literature. We quote from our earlier publication [Kronenberg, Nature Cell Biology 2017]:

“Focal adhesion complexes had sizes of $(6.1 \pm 3.4) \mu\text{m}^2$ and transmitted horizontal forces and stresses of $(3.1 \pm 1.0) \text{ nN}$ and $(602 \pm 323) \text{ Pa}$, respectively, to the substrate ($n=19$ focal adhesions). These values are not significantly different from reference measurements we performed using conventional TFM on stiffness-matched polyacrylamide gels (horizontal stress at focal adhesion, $(386 \pm 131) \text{ Pa}$; $n=6$ cells; $p=0.11$, Mann-Whitney U-test). They are also consistent with earlier micro-pillar array measurements of bovine pulmonary artery smooth muscle cells ($\approx 20 \text{ nN}$ horizontal force for $4 \mu\text{m}^2$ -sized adhesion complexes on an array with fivefold larger effective stiffness)²⁹ and with TFM measurements of mouse embryonic fibroblasts ($\approx 500 \text{ Pa}$ horizontal stress on a hydrogel substrate with a shear modulus of $\approx 2.4 \text{ kPa}$)²⁶.”

26. Legant, W. R. et al. Multidimensional traction force microscopy reveals out-of-plane rotational moments about focal adhesions. Proc. Natl Acad. Sci. USA 110, 881–886 (2013).

29. Tan, J. L. et al. Cells lying on a bed of microneedles: an approach to isolate mechanical force. Proc. Natl Acad. Sci. USA 100, 1484–1489 (2003).

As the fast WARP approach introduced in our current work yields displacement maps that are statistically identical to the much slower ERISM method (Supplementary Note 3), these earlier results are also valid for WARP. However, we have not yet been able to automate the procedure for determining horizontal forces to the same level as we have for vertical forces. We have therefore refrained from including the explicit quantification of horizontal forces in our manuscript. We felt that doing this for a few selected datasets would mislead readers to think that it can be achieved at the high throughput of WARP. Instead, we use the volume by which cells indent into the microcavity as a proxy for the applied force. Prior ERISM studies involving the same microcavity chips have already established that displaced volume can be used as a proxy for force to assess and compare many different types of cellular behaviour. This includes for example the effect of the gene Willin (a key component on the development of Alzheimer’s) on the cellular mechanics (Kronenberg et al. 2020), and the effect of cell-injury on the ability of kidney podocyte cells to exert force (Haley et al. 2018).

To address the referees concern about not determining absolute force values for cardiomyocytes, we have added the following comment to our revised manuscript (lines 212-217):

“Unlike the podosomes analysed above, cardiomyocyte contractions primarily exert horizontal forces on the substrate. We have previously shown that horizontal forces can be quantified by measuring the magnitude of local twisting they induce to the microcavity surface and that focal adhesion forces measured in this way are consistent with other literature reports¹⁷. However, this procedure has required manual image analysis and is thus too time consuming for the multi-frame datasets generated by WARP.”

6. One of the benefits of WARP claimed by the author is that it is “label-free”. However, in practice fluorescent imaging is a very powerful capability for marking different cell types or indicating different biological states. A question is thus whether the WARP cavity which is coated by a gold layer is still compatible with fluorescence imaging or not.

As mentioned above, the WARP technique uses the same microcavity chips for measurements as the prior ERISM studies cited in the manuscript introduction. In these studies, fluorescence measurements have been made in combination with force measurements. This includes, among other examples, invadopodia from head and neck squamous carcinoma [Dalaka, Science Advances 2019] and macrophage podosomes [Kronenberg, Nature Cell Biology 2017], where the identity of the invadopodia and podosomes was confirmed by immunofluorescence. The reason the cavity is compatible with fluorescence imaging is that the gold mirrors have a thickness of only between 10-15 nm. An advantage of WARP and ERISM over fluorescence-based force mapping techniques, such as TFM, is that all spectral channels remain available for fluorescent labelling of cells, while TFM requires one or sometimes two channels for particle tracking.

Reviewer #2 (Remarks to the Author):

The revised paper is much better in conveying the actual physics behind the method, and I appreciate the authors thoughtful responses to the comments. However, I believe that the methodology should be further explained before publication.

1. I feel that the method is the going from the division images to nm displacements is crux of the paper. The demonstrations on cells are fine, and to be honest, no one can really argue with them because part of it has never been measured before. Nevertheless, I think it is imperative that the authors bring in Supp. Note 3 - how to actually get a displacement map into the paper. This is the most important part of the work. Without the mapping between their cotangent to distance, the authors have very little to discuss.

We have now integrated the text from the original Supplementary Note 3 into the main text of the manuscript. This has been included as part of the discussion surrounding Figure 2 on the underlying algorithm for the conversion from interference maps to displacement maps. The original Supplementary Note 3 has been removed from the SI and the references to the sections of the SI updated accordingly.

2. In addition to the comment above, I still feel that a ground truth quantitative comparison is necessary to validate the method.

In addition to the comparison to the displacement maps calculated by the ERISM method from our earlier publication, covered in Fig. 2 h and i and in more detail in the new Supplementary Note 3, we have also now included a “ground truth experiment” where known

forces are exerted on the microcavity by AFM (Supplementary Note 11). As long as the forces are within the linear range of the AFM cantilevers used, the force measured by our method and the force applied by AFM agree to within 10%, which demonstrates the accuracy of the WARP conversion to displacement and the subsequent conversion to stress/force using the FEM software.

REVIEWERS' COMMENTS

Reviewer #1 (Remarks to the Author):

I have studied the revision and the rebuttal and carefully. The manuscript is significantly improved in rigor and the authors sufficiently addressed concerns and limitations raised in previous round of review. I now support its acceptance for publication.

Reviewer #2 (Remarks to the Author):

The decision in the revision to focus on the displacement (rather than forces) was very good. I also appreciate the use FEM to convert to forces. Finally, the double-check with the AFM experiments, is very reassuring for people wary of the accuracy of the method.

I believe the authors have answered my comments and done a reasonable job of answering the other reviewer's questions.

Reviewer #1 (Remarks to the Author):

I have studied the revision and the rebuttal and carefully. The manuscript is significantly improved in rigor and the authors sufficiently addressed concerns and limitations raised in previous round of review. I now support its acceptance for publication.

We thank the referee for taking time to study our revised manuscript and for their positive assessment of our work.

Reviewer #2 (Remarks to the Author):

The decision in the revision to focus on the displacement (rather than forces) was very good. I also appreciate the use FEM to convert to forces. Finally, the double-check with the AFM experiments, is very reassuring for people wary of the accuracy of the method.

I believe the authors have answered my comments and done a reasonable job of answering the other reviewer's questions.

We thank the referee for taking time to study our revised manuscript and for their positive assessment of our work.